# Effects of *Lycium barbarum* L. Polysaccharides on Vascular Retinopathy: An Insight Review

**DOI:** 10.3390/molecules27175628

**Published:** 2022-08-31

**Authors:** Chunhong Yang, Qi Zhao, Shiling Li, Lili Pu, Liqiong Yu, Yaqin Liu, Xianrong Lai

**Affiliations:** 1Department of Chinese Medicine and Pharmacy, College of Pharmacy, Chengdu University of Traditional Chinese Medicine, Chengdu 611137, China; 2Department of Ethnic Medicine, College of Ethnomedicine, Chengdu University of Traditional Chinese Medicine, Chengdu 611137, China

**Keywords:** polysaccharide, retina, bioactivity, mechanism of action, pharmacokinetics, application

## Abstract

Vascular retinopathy is a pathological change in the retina caused by ocular or systemic vascular diseases that can lead to blurred vision and the risk of blindness. *Lycium barbarum* polysaccharides (LBPs) are extracted from the fruit of traditional Chinese medicine, *L. barbarum*. They have strong biological activities, including immune regulation, antioxidation, and neuroprotection, and have been shown to improve vision in numerous studies. At present, there is no systematic literature review of LBPs on vascular retinal prevention and treatment. We review the structural characterization and extraction methods of LBPs, focus on the mechanism and pharmacokinetics of LBPs in improving vascular retinopathy, and discuss the future clinical application and lack of work. LBPs are involved in the regulation of VEGF, Rho/ROCK, PI3K/Akt/mTOR, Nrf2/HO-1, AGEs/RAGE signaling pathways, which can alleviate the occurrence and development of vascular retinal diseases in an inflammatory response, oxidative stress, apoptosis, autophagy, and neuroprotection. LBPs are mainly absorbed by the small intestine and stomach and excreted through urine and feces. Their low bioavailability in vivo has led to the development of novel dosage forms, including multicompartment delivery systems and scaffolds. Data from the literature confirm the medicinal potential of LBPs as a new direction for the prevention and complementary treatment of vascular retinopathy.

## 1. Introduction

With changes in lifestyle and environment, the number of people suffering from various eye diseases reached 710 million in 2019, and the trend is on the rise. It has become a major public health problem worldwide [1]. Vision assumes more than 70% of the external information and is an important sensory organ to maintain the high quality of human life [2]. Retinopathy with progressive and irreversible visual loss or severe visual impairment exists in all age groups. For example, age-related macular degeneration (AMD) is high in the elderly population, and even preterm retinopathy in premature infants, which is at risk of blindness [3]. Therefore, effective prevention and scientific eye treatment have become important concerns of society. Generally speaking, retinal diseases are mainly caused by blood supply disorders, abnormal angiogenesis, fibrosis, hypoxia, etc. [4]. These results suggest that retinopathy is mostly related to vascular abnormalities, which may be a new treatment direction worthy of further exploration.

At present, fundus examination and imaging angiography, and other techniques are commonly used to assist in the treatment of surgery and drugs, including anti-VEGF. However, it is easy to cause surgical complications, including vitreous hemorrhage and has a prognostic risk. At the same time, the economic cost is high, which affects the treatment effect [5]. Therefore, it is more urgent to improve and optimize the treatment methods. Natural plant medicines are rich in source and wide in safety range, which is an important treasure house for finding potential preventive and therapeutic drugs and improving the safety of eye disease indications.

In recent years, polysaccharides as biological response modulators have gradually become a research hotspot. So far, more than 300 natural polysaccharides have been found, which are widely used in medicine, health care, and food [6]. Among them, more than 100 kinds of plant polysaccharides have been proven to play the role of immune regulation [7], anti-tumor activity [8], anti-inflammatory [9], hypoglycemia [10], and anti-viral [11]. For example, ginseng polysaccharides [12] and astragalus polysaccharides [13] have been developed clinically and are widely used. Plant polysaccharides are a class of characteristic biopolymers with low toxicity and low side effects, high biological regulation, and are abundant sources that are composed of many identical or different monosaccharides with α−/ β−glycosidic bonds [14]. LBPs are the main active component of the *L. barbarum* fruit, accounting for about 5–8% of dried fruit, and are also one of the common plant polysaccharides [15]. In addition, *L. barbarum* fruits are rich in substances beneficial to human health, including 0.4% flavonoids, 0.4% flavane-3-alcohols, 1.5% phenolic acids, 0.03% amino acids and derivatives, and 0.1% carotenoids [16]. Their medicinal and nutritional properties are active, and their antioxidant and anti-aging effects have been proven to provide protection for the body [17]. Because of these, *L. barbarum* fruits are widely used as a functional food and medicine for health care and longevity worldwide [18,19]. One of their main traditional applications in China is visual acuity and vision protection [20]. A large number of studies have found that LBPs have a multi-pathway metabolic and highly efficient biological activity in the protection of visual function, which is beneficial to the treatment of vascular retinopathy [16]. So far, there have been no reports on the toxicity and adverse reactions, indicating that it is safe and can be widely used [21].

In fact, the specific role of LBPs in alleviating the development of retinopathy has been gradually confirmed to be important. However, there is no systematic review of these mechanisms. LBPs are a kind of polysaccharide, and their molecular weight and monosaccharide composition ratio will have some differences, which are not consistent. It is difficult to fully connect their physical and chemical characteristics with functional activities. Therefore, the structural characterization and extraction, separation, and purification methods of LBPs were reviewed, which is beneficial to its application research and provides a reference for the study of differences in biological activities. We use “vascular retinopathy,” “*Lycium barbarum* polysaccharides,” and “retina” as the main search terms work through online databases, including Google Scholar, PubMed, Web of Science, and the China National Knowledge Infrastructure Database. The relevant references on LBPs published from the past 12 years (2010–2022) were included. This article mainly summarizes the visual protection characteristics of LBPs and their mechanism regulation in improving common vascular retinopathy and proves that LBPs are a potential preventive and therapeutic agent. We believe that pharmacological studies should pay more attention to the reflection of clinical application. In recent years, reports on LBPs mainly focus on their pharmacological effects but ignore their catabolism in vivo and drug use, which leads to the limitation of their clinical application. Therefore, we also discussed the pharmacokinetics of LBPs and the application of related dosage forms to provide ideas for the application development and drug research of LBPs, with a view to further in-depth research.

## 2. Bioinformatics of LBPs

*Lycium barbarum* L. (belonging to Solanaceae) is widely distributed in the arid to semi-arid regions of North and South America, Africa, and Eurasia [22]. China is the main production area, among which Ningxia is recorded as the original production area [23]. Its fruit is high in nutrition and has a healthy biological activity, and has a sweet taste, which has attracted worldwide attention to it as a “superfood” with increasing demand [24]. *Lycium barbarum* L. is a traditional medicinal material and functional tonic widely used in Asian countries as a medicinal and edible plant. Especially in China, its function in eyesight has been clearly recorded in various ancient medical books, including the Compendium of Materia Medica, which has been used for centuries [17]. LBPs are a group of water-soluble proteoglycans isolated from it, and its research is mainly focused on crude polysaccharides.

The results of ultra-high-performance liquid chromatography quadrupole trap tandem mass spectrometry (UHPLC-QTRAP-MS/MS) showed that the LBP was composed of galactose, arabinose, mannose, rhamnose, xylose, ribose, and glucose, among which glucose was the most abundant monosaccharide [25]. As shown in Figure 1, (1→3) -β-D-pyranogalactosyl, (1→4) -α-D-pyranogalactosyl residues, and (1→6) -β-D-pyranogalactosyl are three common characteristic structures [24]. The special structure of natural polysaccharides, including the molecular weight, monosaccharide composition, glycosidic bond, and charge characteristics, determines the biological activity of polysaccharides [26,27]. However, the physicochemical properties of polysaccharides were affected by the extraction and purification methods. Therefore, it is necessary to understand the extraction and transmission process of LBPs. We briefly review the extraction, isolation, and purification methods that have been used for LBPs, as shown in Figure 2. The common extraction methods include hot water extraction, microwave-assisted extraction (MAE), ultrasonic-assisted extraction (UAE), and pressurized liquid extraction (PLE) [28]. Protein is the main impurity in the extraction process of LBPs. Alcohol precipitation methods, including graded alcohol precipitation [29] and membrane separation [30], are usually used, but the yield is low, and only one method is usually difficult to obtain the ideal pure polysaccharide [26]. Therefore, some efficient methods have been applied to study the ethylene oxide-b-propylene-b-ethylene oxide (EOPOEO) [31] and column chromatography [27]. A variety of methods have been combined to improve the separation and purification methods, and good results have been achieved. All of these provide a reference for the functional application of the LBP, which is conducive to further exploring the comparative study of biological activities. 

In addition, the pharmacological properties of LBPs include antioxidant [32], anti-inflammatory [33], nerve protection [34], and immunity [35]. They play an important role in the signaling pathway involving Nrf2/HO-1, PI3K-AKT-mTOR, and P38-MAPK [35]. As a good candidate for prevention and treatment, LBPs have been found to be used to improve the gut microbiota to treat fatty liver disease [36], participate in the Nrf2/HO-1 mechanism to prevent neurodegenerative diseases [37], enhance cell viability through immunity and anti-oxidation to act on a variety of eye diseases [38], and regulate the gut microbiota to improve hyperlipidemia [39]. At present, a large number of studies have predicted that LBPs have positive-targeted therapeutic effects on retinopathy.

## 3. LBPs and Vascular Retinopathy

The pathological mechanism of vascular retinopathy is the destruction of the protective barrier and the balance level of related factors, which is manifested as angiogenesis, endothelial cell dysfunction, nerve cell damage, and so on [4,40,41]. In recent years, a large amount of literature has proved that LBPs can be used in the treatment of vascular retinopathy, including diabetic retinopathy (DR), AMD, hypertensive retinopathy, and ischemic/reperfusion (I/R) injury [32]. This paper summarizes the protective effects of LBPs on the vascular retina in vivo and in vitro model experiments, as shown in Table 1. The retina has a unique and complex physiological structure. Understanding the important parts of the retina can contribute to a better understanding of the prevention and treatment of retinal diseases. Vascular retinopathy mainly occurs in the blood–retinal barrier and neuroglia, and the blood vessels are wrapped in the innermost layer, followed by the internal blood-retinal barrier (iBRB), the external blood–retinal barrier, and the glial cells, as shown in Figure 3 [42,43,44,45,46]. Different locations of lesions affect the different occurrences of lesions, so we will elaborate on the specific effects of them in the following sections.

### 3.1. Diabetes Retinopathy

DR is a microvascular complication of endothelial dysfunction caused by diabetes. The main pathological features are a blood–retinal barrier (BRB), retinal thinning, increased neovascularization, and the degeneration of neurons [58,59]. The molecular mechanism is still unclear.

In recent years, proliferative DR characterized by pathological retinal neovascularization has gradually become one of the major complications of visual threat [60]. The vascular endothelial growth factor (VEGF) is a key inducer in stimulating the abnormal growth of new vessels in DR and is involved in multiple pathways, including oxidative stress and apoptosis, while the pigment epithelium-derived factor (PEDF), as a protective factor, can inhibit angiogenesis [61]. Retinal angiogenesis depends on the balance between the pro-angiogenic factor (VEGF) and the anti-angiogenic factor (PEDF) [62]. In DR, the overexpression of the VEGF leads to retinal vascular abnormalities. The LBP can enhance the expression of the PEDF and inhibit the strong activity of the VEGF and regulate the influence of both levels to restore the equilibrium state and reduce the formation of new blood vessels [47]. In addition, miR-15a is considered to be a key regulator of the pro-angiogenesis pathway. On the one hand, it directly affects the VEGFR and interferes with the VEGF expression, while, on the other hand, it reduces acid sphingomyelinase (ASM) in endothelial cells and inhibits angiogenesis in high glucose environments [63,64]. As a biomarker for predicting type 2 diabetes, it may be a potential therapeutic target for DR [65]. The overexpression of miR-15a can enhance the activity of tight junction proteins, Zonula Occludens-1 (ZO-1) and Occludin, inhibit TGFbeta3/VEGF signal transduction, and maintain the normal function of retinal endothelial cells [66]. It was reported that the LBP down-regulated the expression of miR-15a-5P in monkey retinal vascular endothelial (RF/6A) cells under high glucose conditions to rescue retinal angiogenesis [48].

Studies have shown that the LBP has a positive protective effect on the BRB. The BRB determines visual function and maintains the homeostasis of the retinal microenvironment by selectively regulating the molecular flux and substance exchange between the blood and retina [67]. The internal BRB (iBRB) are tightly connected by blood vessels and endothelial cells, and their functional properties depend on the molecular activity of tight junction proteins, including ZO-1/-2/-3, cadherin-5, cingulin, occluding, symplekin, claudins, and so on [68,69,70]. However, hyperglycemia can promote morphological changes in human retinal microvascular endothelial cells by transforming the growth factor -β1 (TGF-β1) and pro-inflammatory cytokines (IL-1β and TNF-α). After exposure, the endothelium-interstitial transformation occurs, resulting in abnormal endothelial cell function and reducing the expression of markers [71]. Studies have shown that iBRB destruction is a trigger for various eye diseases characterized by retinal edema and vascular leakage [72,73]. LBPs protect the iBRB by regulating the Rho/ROCK signaling pathway. This pathway can regulate the expression and function of the intercellular adhesion molecule-1 (ICAM-1) in endothelial cells, block endothelial hyperpermeability induced by the VEGF, and improve endothelial cell injury and vascular leakage [74,75]. In the STZ-induced diabetic rat model, LBP reversed the ROCK activation state of the endothelial cells, inhibited the activities of ROCK and P-MLC, and up-regulated the expression of P-Occludin to alleviate retinal damage [49]. 

In conclusion, LBPs protect the iBRB and endothelial cells in the improvement of DR mainly by regulating the levels of the VEGF, miR-15a factor, and related upper and lower signal molecules in the Rho/ROCK signal pathway. The related mechanisms are shown in Figure 4.

### 3.2. Age-Related Macular Degeneration

AMD occurs in the macular area of the retina and is characterized by the progressive loss of central vision. It is an important cause of vision problems in an aging population [68]. AMD often occurs in the external BRB, and its structural integrity is supported by the choroid, Bruch’s membrane (BM), and retinal pigment epithelium (RPE) cells [76,77]. The prominent pathological features of early AMD are drusen and RPE abnormality. RPE is a potential cell source for retinal neuron regeneration, which is beneficial for maintaining retinal function [78]. Therefore, timely intervention in the development of AMD is particularly important.

Although the mechanism of apoptosis in the pathological process of AMD has not been elucidated, the abnormal accumulation of all-trans retina is associated with caspase-3/ Gasdermin E-mediated scortosis [79]. Apoptosis begins with activation of the inflammasome, including NLRP3 and Procaspase-1 (precursors), and so on. Caspase-1 helps pro-inflammatory cells secrete IL-1β and Il-18 to trigger inflammation, and the cleavage of GSDMD leads to cell lysis and swelling [80,81]. LBPs can reduce the inflammatory response and regulate autophagy, which has a neuroprotective effect [50]. In the light-induced apoptosis experiment of RPE cells, LBPs significantly enhanced cell viability and up-regulated the phosphorylation levels of Akt and mTOR proteins, thereby activating the PI3K/Akt/mTOR signaling pathway. It can accelerate the proliferation and migration of retinal endothelial cells and inhibit the excessive autophagy of RPE cells [51]. The balance of the pro-apoptotic gene, Bax, and the anti-apoptotic gene, Bcl-2, in vivo, critically regulates the process of apoptosis. The LBPs inhibited apoptosis, enhanced antioxidant enzymes, and activated the Nrf2/HO-1 pathway by down-regulating pro-apoptotic proteins (Bax and Caspase-3) and increasing the Bcl-2 expression, alleviating the oxidative damage of ARPE-19 cells induced by H2O2 [52,82,83]. In addition, Aβ 1–40 oligomers are mainly concentrated in drusen, possibly activating the NLRP3 inflammasome and oxidative stress pathways. At the same time, exposure to the Aβ 1–40 oligomer can activate pyrophosis of the ARPE-19 cells, resulting in the decreased activity of the RPE cells [84]. In fact, LBPs can disrupt the oligomerization of Aβ 1–40 and inhibit the activation of apoptosis, reducing the release of NLRP3, caspase-1, GSDMD-N, as well as IL-1β and IL-18, which has been demonstrated in the treatment of liver injury [85] and lung injury [86] by LBPs [53]. It has also been confirmed that LBPs can regulate the production of the β-amyloid protein, which is conducive to the maintenance of the BRB and has a potential therapeutic effect on mediating vascular retinal diseases [54].

In conclusion, LBPs enhance cell viability and delay the progression of AMD through anti-inflammatory, apoptosis, and autophagy pathways, and regulates the PI3K/Akt/mTOR pathway.

### 3.3. Hypertensive Retinopathy

Hypertension can improve systemic arterial pressure and peripheral vascular resistance and is the main risk factor for systemic vascular diseases. The fundus changes caused by it include arteriole stenosis, cotton patch, retinal hemorrhage, papilledema, and other microvascular and optic nerve abnormalities, with high intraocular pressure as the main manifestation [87,88]. 

In hypertensive retinopathy, the glial fibrillary acid protein (GFAP) expression is significantly increased, suggesting that it is accompanied by a neuroinflammatory response. At present, the key regulatory role of Müller glial cells in the inflammatory response has attracted much attention [89,90]. Astrocytes are closely related to angiogenesis and play an important role in neurovascular homeostasis. The GFAP has been found to be useful in assessing astrocyte activation levels in previous studies [91]. It was found that LBPs can protect neurons and blood vessels in acute/chronic ocular hypertension animal models. Microglia are the main immunocompetent cells in the neurovascular system. Overactivation can secrete harmful factors, including ROS and pro-inflammatory cytokines, and play an important role in the pathogenesis of retinal ganglion cells (RGC) [92]. LBPs can protect neural function by moderately activating microglia or inhibiting the activation of the NLRP3 inflammasome from regulating autophagy and the MAPK pathways to improve microglia damage [93,94]. RAGE can activate the membrane transport system of AGE-RAGE and Aβ [95]. The LBPs decreased the expression of Aβ and AGEs, down-regulated the GFAP, and mediated the activity of the retinal glial cells to improve the BRB and neuron survival. The overexpression of the vasoconstrictor endothelin-1 (ET-1) down-regulated Occludin levels, resulting in BRB damage, which was released after the activation of the RAGE and Aβ membrane transport system. The LBPs inhibited vascular injury and the degeneration of RGC neurons in AOH injury by down-regulating RAGE and ET-1-related signaling pathways in the retina [53,96].

In conclusion, LBPs regulate the AGEs/RAGE pathway and ET-1 factor mainly through apoptosis, autophagy, and neuroprotection in the improvement of hypertensive retinopathy.

### 3.4. Ischemia-Reperfusion Injury

Retinal I/R injury is a common complication of various metabolic disorders, including BRB destruction, glial cell activation, oxidative stress, and neuronal death [97].

The Nrf2/ HO-1 pathway is one of the main antioxidant ways involved in weakening the oxidative stress response and maintaining the redox state in many tissues. Heme oxygenase 1 (HO-1) has a large number of antioxidant elements, making it a potential therapeutic target for the prevention of neurodegenerative diseases. Nrf2 showed a high protective effect on neuron and vascular degeneration in retinal I/R injury [98]. Nrf2 releases and activates HO-1 in response to ROS stimulation. LBPs can activate the Nrf2/HO-1 antioxidant pathway and reduce ROS production to protect the retina [56]. In addition, ROS produced by ischemia increases vascular permeability by up-regulating the VEGF gene expression. The overexpression of aquaporin 4 (AQP4) leads to retinal swelling and BRB tight junction injury. The LBPs exerted neuroprotective effects by down-regulating AQP4 activity, maintaining the BRB’s structural integrity, and ameliorating characteristic ocular diseases about I/R [99].

Although there are few direct studies on the LBP, it can be used as a potential therapeutic mechanism. Previous studies have shown that the immune response can affect the degeneration process of neurons in the central nervous system [57,100]. In fact, it has long been found that LBPs can enhance immune function, and it is speculated that the beneficial effect of LBPs on improving I/R may also be mediated by immune regulation, but there is no conclusion at present [101,102,103,104].

According to the summary of the above mechanism of action, we can clearly know that LBPs can improve the occurrence and development of vascular retinopathy through multiple metabolic pathways, including inflammatory response, oxidative stress, apoptosis, autophagy, and neuroprotection, as shown in Figure 5. This also suggests that LBPs are abundant in physiological activity and may serve as a potential candidate for prevention and treatment.

## 4. Pharmacokinetics

In recent years, the therapeutic potential of LBPs has attracted extensive attention from scholars. The metabolic analysis of the LBP in vivo is helpful to understand its target, but there are few studies on it. Fluorescein isothiocyanate (FITC) can be highly reactive with free amino modification in LBPs. Now, it is widely used for quantitative determination by fluorescent labeling [105].

Studies have found that the LBP is absorbed quickly in the body, mainly in the small intestine and stomach. Overall, the LBP-FITC content in the heart, small intestine, and plasma decreased with time. The concentration of the liver, kidney, and large intestine reached the highest after 6 h, while the concentration of the large intestine reached the highest after 24 h. After 1 h of administration to rats, the concentration of the LBP-FITC was mainly in the small intestine and stomach, followed by the liver, large intestine, heart, and kidney, which were all higher than in the plasma. After 6 h, the content of the LBP-FITC in the kidney was the highest and increased in the large intestine and liver. After 24 h, the LBP-FITC content decreased in other tissues but increased in the muscles [106]. In addition, the absorption rate of the LBP was the highest in the duodenum, followed by the jejunum and ileum, but the overall absorption rate of these three intestinal parts was low, and not more than 15% [107]. However, there are different reports that LBP’s molecular weight and monosaccharide composition do not change significantly during simulated gastrointestinal digestion in vitro, which does not appear to be degraded [108].

Studies have proved that the LBP is not highly absorbed in the body and is eliminated slowly, and the LBP-FITC is mainly excreted through the urine and feces in the prototype. After 72 h, the excretion rate of the LBP-FITC in the urine and feces accumulated was 92.27%, of which the urine excretion was only 0.09% [109]. This indicates that a large amount of the LBP-FITC is excreted, and the absorption in the body is not ideal.

In summary, according to existing reports, the LBP is quickly absorbed in the body but not easily absorbed, and its elimination rate is slow. Direct oral administration may act on the intestine. An in vitro simulation experiment also confirmed that LBPs utilized clathrin-mediated endocytosis to transport monolayers through Caco-2 cells, suggesting that LBPs could be absorbed by the intestine [110]. Currently, the pharmacokinetic studies of LBPs are still limited, which is insufficient to fully explain the in vivo active information. Therefore, further studies on the metabolic level are needed.

## 5. Application Prospect

Due to the uniqueness of the polysaccharide macromolecular structure and ocular physiological structure, its dosage form is an important factor in the effective prevention and treatment of diseases. LBPs show active biological activity in vascular retinopathy and have potential as a drug for prevention and treatment. However, the bioavailability of LBPs is low after direct oral administration. How to exert efficient pharmacological effects in vivo is an important problem to be solved at present. We briefly summarize the recent years of LBPs’ derivatization of new drug dosage forms to provide a reference for future drug development.

In recent years, the development of neural tissue engineering has brought the hope of nerve regeneration. A scaffold made of electrospun nanofibers can play the function of the extracellular matrix, guide axonal repair after nerve injury, and deliver drug molecules and cytokines, which has broad application prospects in clinical preparations [111,112]. Polylactic acid-glycolic acid copolymer (PLGA) is a potential candidate due to its biodegradability and biocompatibility [113]. An experiment demonstrated that LBP-PLGA nanofiber scaffolds could promote Schwann cell differentiation, PC12 cell proliferation, and neuronal axon growth [114]. In addition, the LBP is often used in the treatment of diseases by multi-compartment drug delivery systems, including liposomes and nanoparticles as dosage forms. LBP liposomes activate dendritic cells and act as immune adjuvants to stimulate the secretion of macrophages and T cytokines, improve immune activity, and regulate immune response [115,116]. 

LBPs have little research on the drug delivery system for retinal diseases, and their application is limited. The methods of ocular drug delivery methods include common ophthalmic liquid/solid preparations, including gel [117], nanoparticles [118], lyophilized powder, ocular inserts, and medical contact lenses [119], which may become the direction of LBPs’ preparation development and can be actively promoted for further application research. However, the problems of drug residence time and drug permeability in the eye still need to be further explored.

## 6. Conclusions

LBPs are a group of natural proteoglycans. The extraction method of LBPs affects the structural properties, which in turn affects the biological activity, but their correlation and differences need to be further studied. LBPs ameliorate vascular retinopathy by protecting the blood–retinal barrier and neuroglia cells. The experimental data showed that LBPs involved in the VEGF, mir-15a, and Rho/ROCK signaling pathway associated with the level of upper and lower signaling molecules to reduce DR. LBPs could improve the viability of RPE cells by regulating Bax/Bcl-2, PI3K/Akt/mTOR, and Nrf2/HO-1 pathways, acting on anti-inflammatory, apoptosis, and autophagy. LBPs mainly regulate the AGE-RAGE pathway, ET-1, and GFAP signal levels, participate in apoptosis, autophagy, and immune pathways, protect nerve cells and delay the progression of hypertensive retinopathy. The slowing down of I/R progression was mainly attributed to an immune response and the Nrf2/HO-1 antioxidant pathway. Through its role in vascular protection, LBPs will also become a potential treatment and supplement for vascular-related retinopathy. In addition, the absorption of LBPs in the body is fast, mainly through the small intestine and stomach, and the elimination rate is slow through urine and stool excretion, which provides experimental data support for its further study.

In conclusion, LBPs have been supported by multiple data as a potential candidate for the prevention and treatment of vascular retinopathy, and it is promising for drug development, but further preclinical studies are needed. This review focuses on summarizing the protective effect of LBPs on vascular retinopathy, proving the potential of treatment and application. It is beneficial to targeted therapy research, providing basic support for further clinical application and hoping to promote the application research of natural products.

## Figures and Tables

**Figure 1 molecules-27-05628-f001:**
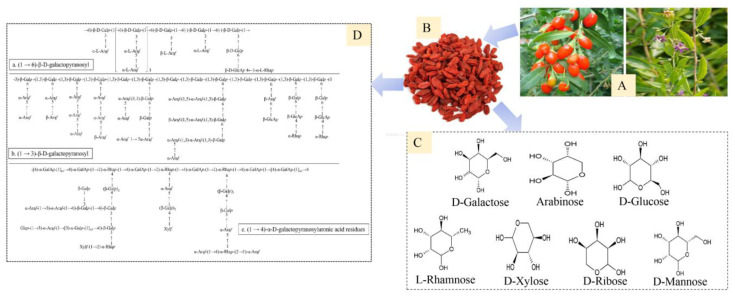
The characterization structure of LBPs. (**A)** the fresh fruit of *L. barbarum*; (**B)** the dried fruit of *L. barbarum*; (**C)** the monosaccharide composition in LBPs; (**D)** three representative structures of LBPs, including (1→6)-β-D-pyranogalactosyl, (1→3)-β-D-pyranogalactosyl, and (1→4)-α-D-pyranogalactosyl residues.

**Figure 2 molecules-27-05628-f002:**
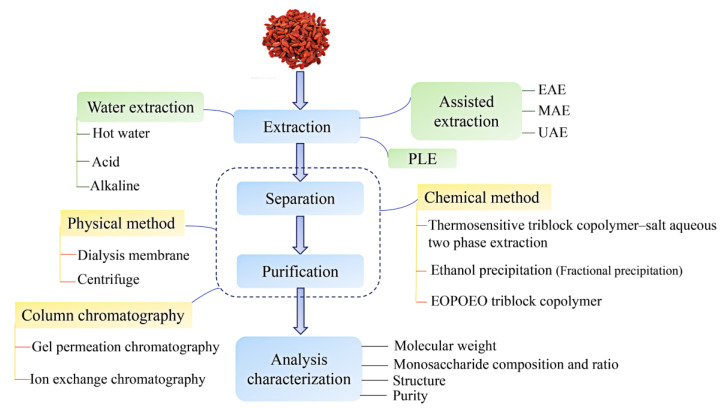
The common methods of extraction, separation, and purification of LBPs.

**Figure 3 molecules-27-05628-f003:**
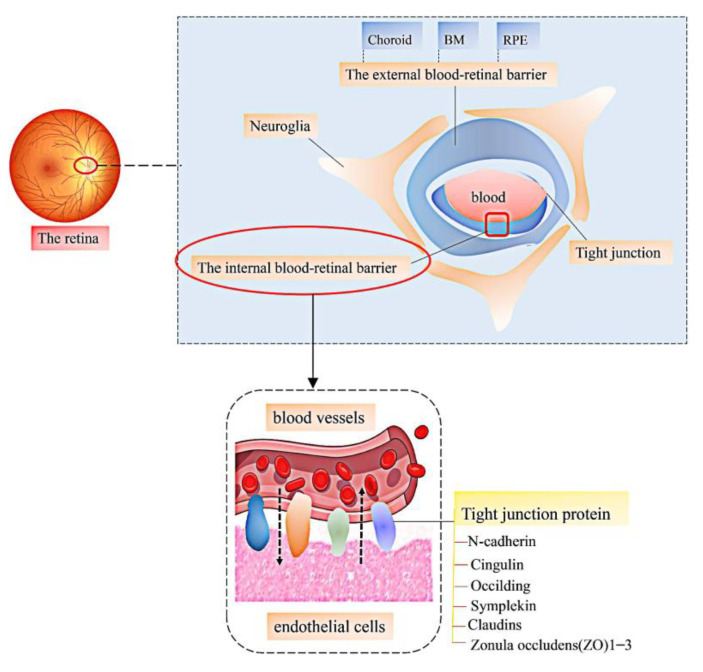
The main pathogenic sites in vascular retinopathy. The physiological connections between the human retina and blood vessels are mainly internal and external blood–retinal barriers, glia, etc. The internal blood–retinal barrier is an important part, and a tight junction protein is the key part of it.

**Figure 4 molecules-27-05628-f004:**
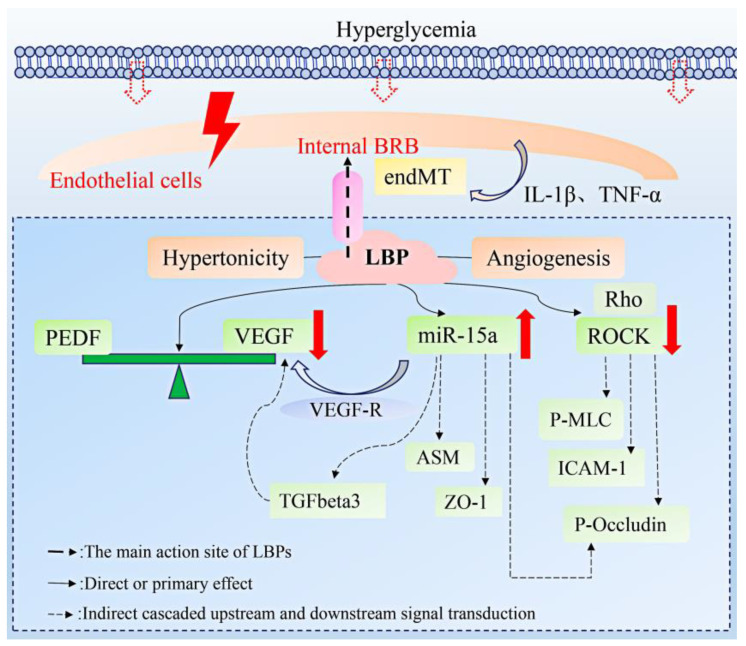
The mechanism of action of LBPs to improve diabetic retinopathy.

**Figure 5 molecules-27-05628-f005:**
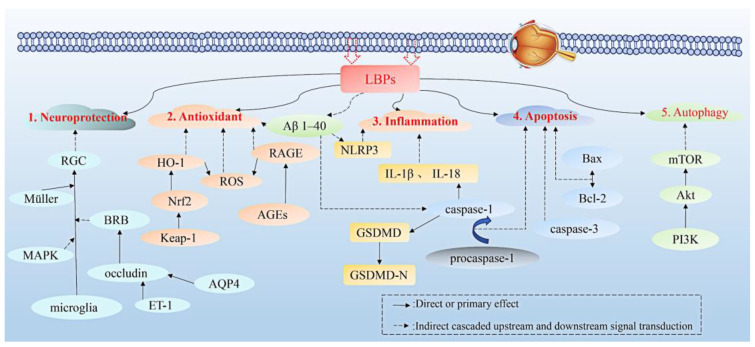
The pathway of LBPs in improving vascular retinopathy. The protective effects of LBPs are mainly attributed to neuroprotection, antioxidant, inflammation, apoptosis, and autophagy.

**Table 1 molecules-27-05628-t001:** Summary of protective effect of the LBP on vascular retinopathy in the experimental model.

The Experimental Model	Dosage of Administration	Application	Mechanism/Pathway (Major Molecular Change)	Reference
STZ induced -Male Sprague-Dawley rats (250 ± 20 g)	200, 400 mg/kg/d orally for 20 weeks	in vivo	Decreasing the immune intensity of GFAP and VEGF overexpression, increasing PEDF expression	[47]
Monkey retinal vascular endothelial (RF/6A) cells	600 mg/L for 48 h	in vitro	Decreasing VEGFA, VEGFR2, ANG2, ASM mRNA, and protein expression while increasing ANG1 protein expression	[48]
STZ induced-diabetic rat (8–12 weeks, 180–220 g)	250 mg/kg/d for 12 weeks	in vivo	Increasing P-occludin, down-regulating ROCK1, and P-MLC	[49]
BV2 cells	300 μg/mL	in vitro	Significantly reducing NLRP3, cleaved caspase-1, IL-1β, IL-18, and P62	[50]
ARPE-19 cells	10, 50, 100 mg/L for 24 h	in vitro	Increasing PI3K, P-mTOR/mTOR, and P-Akt/Akt levels	[51]
ARPE-19 cells	0, 0.25, 0.5, 1 and 2 mg/mL for 2 h	in vitro	Decreasing ROS production of alleviating OxS increased Nrf2 nuclear translocation and HO-1 expression	[52]
Male C57BL/6N mice	1 mg/kg/d for 7 days	in vivo	Decreasing the Aβ level of RAGE expression in RGC	[53]
N2a/APP695 cells	0, 1.25, 2.5 and 5 μM for 24 h	in vitro	Decreasing in the ratio of Aβ42/Aβ40 in N2a/APP695 cells to protect nerves from damage	[54]
Adult female Sprague-Dawley rats (10-week-old,180–200 g)	1, 10 mg/kg/d for 28 days	in vivo	Significantly reducing retinal inner thickness (IRLT) and positive dark threshold response (pSTR), inhibiting secondary degeneration	[55]
male Sprague-Dawley rats (8 weeks, 300–350 g)	1 mg/kg/d for 1 week	in vivo	Inhibiting RGC loss and ROS production and enhancing Nrf2 and HO-1 immune reaction activities	[56]
C57BL/6N male mice (10–12 weeks old)	1 mg/kg/d for 1 week	in vivo	Decreasing activation of GFAP and AQP4 and down-regulating levels of IgG exosmosis and PAR expression	[57]

## Data Availability

Not applicable.

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
