# Peer review of "Effects of *Lycium barbarum* L. Polysaccharides on Vascular Retinopathy: An Insight Review"

_molecules, 2022, doi:10.3390/molecules27175628_

Round 1
Reviewer 1 Report
Journal Molecules (ISSN 1420-3049)
Manuscript ID molecules-1863082
Type Review
Title Protective effects of Lycium barbarum polysaccharide on vascular retinopathy
Authors Chunhong Yang , Qi Zhao , Shiling Li , Lili Pu , Liqiong Yu , Yaqin Liu , Xianrong Lai *
Dear Editor,
Thank you for inviting me to review the manuscript “Protective effects of Lycium barbarum polysaccharide on vascular retinopathy.”
The manuscript addresses an important issue of vascular retinopathy. However, it requires substantial revision and additional information. Below are some comments that will improve the quality of this manuscript.
Kind regards
Title
1. Latin names of the species should be written in italics and the discoverer’s name should be provided - “Lycium barbarum L.”
2. The title should inform that this is a review paper – “A review”
Keywords
3. Eliminate words that are used in the title.
Abstract
4. Delete lines 10 and 11.
5. Latin names of the species should be italicised (throughout the manuscript)
6. Eliminate the expression “…such as …” in l. 18, 36, 68, 76, 116, 169, 204, 216, 295, 299, 304, 307
7. l. 12, 13. Explain the change and give exact information on the polysaccharide in “Lycium barbarum polysaccharide” (this expression conveys non-specific information).
8. l. 13, 14 Explain the mechanism of the biological activity.
9. Give precise information pertaining to the topic of the manuscript - “At present, there is no systematic literature review related to LBP.” (there are many publications on L. barbarum polysaccharides exclusively)
8. Specify the year range of the analysed literature.
9. Eliminate mental shortcuts – “other published resources”; instead, provide information on the databases or sources w
10. The Abstract should present specific conclusions formulated based on the analysis of many original research publications.
Introduction
11. Please complete the Introduction by mentioning the latest literature reports of original research publications related to the topic of the study. The 7 sentences based on 5 literature references cannot constitute an "Introduction" section in an international journal with high impact factors. I suggest introducing the subject of the manuscript to the reader by presenting a general outline of the studied issues e.g.
(i) the raw material used in various experimental models in relation to the topic of the study
(ii) the main biologically active chemical compounds (instead of the class of "polysaccharides") contained in the raw material relevant to the topic mentioned
(iii) metabolic mechanisms
(iv) research on cell lines
(v) animal studies
(vi) clinical trials, etc.
12. l. 49 – 56 Provide the literature reference.
13. l. 47 – 48 The sentence is general and does not provide substantive information.
14. Specify the methods and the group of drugs in “…optimize the treatment methods and to find potentially valuable drugs to improve the safety of eye disease indications…”
l. 49-61
15. Indicate which raw material is used “…as a functional food….”
16. What is the mechanism of action of the raw material as “medicine for health care”
17. Explain what mechanisms guarantee “…longevity all over the world.”
“Lycium barbarum L. is widely used as a functional food and medicine for health care and longevity all over the world.”
18. Please indicate biologically active compounds related to the topic of study.
19. The statement “Lycium barbarum polysaccharide…” - is a broad term that includes various chemical compounds.
20. Please specify the rationale of the undertaken research issue
21. Formulate the aim of the study clearly.
Materials and Methods
22. See comments 7 and 8.
23. Complete the information on the scheme of analysis of the collected publications.
24. Provide information on the assumed theses in the analysis of the original research publications.
Results
25. Please check the structure (layout) of the manuscript in accordance with the requirements for authors for this type of publications.
26. l. 73-77. Provide information on the raw material used.
27. l.81 – 82 Is anything missing in the text???„Generally, the chemical properties are evaluated by the content of total polysaccharides, 81 uronic acid and protein [7].”
28. l.82 – 84 Explain the abbreviation “… UHPLC‐QTRAP‐MS/MS)…”
29. l.84-91 Explain the mechanism of action in “…pharmacological effects…” in relation to the topic of the study.
30. Each activity should be supported by relevant research publications. “(anti-oxi-85 dation, anti-inflammatory, nerve protection, immune “
31. Explain the abbreviations of …Nrf2/HO-1, PI3K-Akt-mTOR, P38-MAPK and other signalling pathways.
32. Specify the pathways in the expression “…and other signaling pathways [12].”
33. Provide information on the mechanism of action in the phytotherapy of the disease entities mentioned. “As a good candidate for prevention and treatment, LBP has been found to 88 treat fatty liver disease [13], neurodegenerative diseases [14], a variety of eye diseases [15], 89 hyperlipidemia [16], etc.”
34. l. 91 – 93 figure 1
35. Provide good quality photographs.
36. Improve the quality of schematic formulas and captions.
37. Provide the correct description of figure 1 to be in line with what it presents.
38. l. 94 – 100 Add a literature reference.
39. L. 105 – 106 Table 1.
40. Italicise “in vivo” and “in vitro”.
41. Explain all abbreviations.
Figure 2.
42. Correct the quality of figure 2 (as well as other figures).
43. Provide information on future research.
44. Indicate a group of readers who will find the present literature review useful.
The entire text requires detailed revision in terms of thematic consistency and supplementation of the quotation of the original scientific literature (it is impossible to mention all shortcomings).
Conclusion and Prospect
45. In line with the instructions for authors, pay attention to the structure of “Conclusion and Prospect” ???
46. The conclusions should be a response to the aim of the study and assumed theses.
47. Please formulate specific conclusions following from the analysis of the publications. Present other information related to the subject of the manuscript in separate subsections.
48. The Conclusion section should not contain literature references.
References
49. In accordance with the instructions for authors, check the following elements:
(i) the use of a comma or a colon in the information about pages (references 12 and 14)
(ii) italics in Latin species names or ex vivo (references 5 and 6)
(iii) check the spaces (reference 11)
(iv) correct (references 2 and 4 etc.)

Author Response
Dear Editors and Reviewers,
We appreciate your professional comments concerning our manuscript entitled “Protective effects of Lycium barbarum polysaccharide on vascular retinopathy”. These kind comments are all valuable and very helpful in revising and improving the quality of the article.
We have studied the comments carefully and have made corrections which we hope meet with approval. We hope that the refined manuscript and supplementary materials will meet the requirements of the reviewers and the high standards of publication. Please use the revision mode to track the changes in the article. And, we have carefully replied to the comments of the reviewer. Please see the attachment.
Kind regards

Reviewer 2 Report
1. Table 1: Authors have reported similar findings at different doses. Then What is the rationale to use a different dose?
2. Figure resolution and presentation are poor.
3. Authors have used abbreviations in the manuscript. A detailed list is required.
4. Chemistry part must be added in the manuscript about the plant.
5. Author must add the delivery system used for this plant extract.
Author Response

(The authors gave the same response as above.)

Round 2
Reviewer 2 Report
accept
Author Response
感谢您的宝贵意见!